# Ancestral Causal Inference

**Sara Magliacane**
VU Amsterdam & University of Amsterdam
sara.magliacane@gmail.com

**Tom Claassen**
Radboud University Nijmegen
tomc@cs.ru.nl

**Joris M. Mooij**
University of Amsterdam
j.m.mooij@uva.nl

## Abstract

Constraint-based causal discovery from limited data is a notoriously difficult challenge due to the many borderline independence test decisions. Several approaches to improve the reliability of the predictions by exploiting redundancy in the independence information have been proposed recently. Though promising, existing approaches can still be greatly improved in terms of accuracy and scalability. We present a novel method that reduces the combinatorial explosion of the search space by using a more coarse-grained representation of causal information, drastically reducing computation time. Additionally, we propose a method to score causal predictions based on their confidence. Crucially, our implementation also allows one to easily combine observational and interventional data and to incorporate various types of available background knowledge. We prove soundness and asymptotic consistency of our method and demonstrate that it can outperform the state-of-the-art on synthetic data, achieving a speedup of several orders of magnitude. We illustrate its practical feasibility by applying it to a challenging protein data set.

## 1 Introduction

Discovering causal relations from data is at the foundation of the scientific method. Traditionally, cause-effect relations have been recovered from experimental data in which the variable of interest is perturbed, but seminal work like the *do*-calculus [16] and the PC/FCI algorithms [23, 26] demonstrate that, under certain assumptions (e.g., the well-known *Causal Markov* and *Faithfulness* assumptions [23]), it is already possible to obtain substantial causal information by using only observational data.

Recently, there have been several proposals for combining observational and experimental data to discover causal relations. These causal discovery methods are usually divided into two categories: constraint-based and score-based methods. Score-based methods typically evaluate models using a penalized likelihood score, while constraint-based methods use statistical independences to express constraints over possible causal models. The advantages of constraint-based over score-based methods are the ability to handle latent confounders and selection bias naturally, and that there is no need for parametric modeling assumptions. Additionally, constraint-based methods expressed in *logic* [2, 3, 25, 8] allow for an easy integration of background knowledge, which is not trivial even for simple cases in approaches that are not based on logic [1].

Two major disadvantages of traditional constraint-based methods are: (i) vulnerability to errors in statistical independence test results, which are quite common in real-world applications, (ii) no ranking or estimation of the confidence in the causal predictions. Several approaches address the first issue and improve the reliability of constraint-based methods by exploiting redundancy in the independence information [3, 8, 25]. The idea is to assign weights to the input statements that reflect

their reliability, and then use a reasoning scheme that takes these weights into account. Several weighting schemes can be defined, from simple ways to attach weights to single independence statements [8], to more complicated schemes to obtain weights for combinations of independence statements [25, 3]. Unfortunately, these approaches have to sacrifice either accuracy by using a greedy method [3, 25], or scalability by formulating a discrete optimization problem on a super-exponentially large search space [8]. Additionally, the confidence estimation issue is addressed only in limited cases [17].

We propose Ancestral Causal Inference (ACI), a logic-based method that provides comparable accuracy to the best state-of-the-art constraint-based methods (e.g., [8]) for causal systems with latent variables without feedback, but improves on their scalability by using a more coarse-grained representation of causal information. Instead of representing all possible direct causal relations, in ACI we represent and reason only with ancestral relations ("indirect" causal relations), developing specialised ancestral reasoning rules. This representation, though still super-exponentially large, drastically reduces computation time. Moreover, it turns out to be very convenient, because in real-world applications the distinction between direct causal relations and ancestral relations is not always clear or necessary. Given the estimated ancestral relations, the estimation can be refined to direct causal relations by constraining standard methods to a smaller search space, if necessary.

Furthermore, we propose a method to score predictions according to their confidence. The confidence score can be thought of as an approximation to the marginal probability of an ancestral relation. Scoring predictions enables one to rank them according to their reliability, allowing for higher accuracy. This is very important for practical applications, as the low reliability of the predictions of constraint-based methods has been a major impediment to their wide-spread use.

We prove soundness and asymptotic consistency under mild conditions on the statistical tests for ACI and our scoring method. We show that ACI outperforms standard methods, like bootstrapped FCI and CFCI, in terms of accuracy, and achieves a speedup of several orders of magnitude over [8] on a synthetic dataset. We illustrate its practical feasibility by applying it to a challenging protein data set [21] that so far had only been addressed with score-based methods and observe that it successfully recovers from faithfulness violations. In this context, we showcase the flexibility of logic-based approaches by introducing weighted ancestral relation constraints that we obtain from a combination of observational and interventional data, and show that they substantially increase the reliability of the predictions. Finally, we provide an open-source version of our algorithms and the evaluation framework, which can be easily extended, at `http://github.com/caus-am/aci`.

## 2   Preliminaries and related work

**Preliminaries**   We assume that the data generating process can be modeled by a causal Directed Acyclic Graph (DAG) that may contain latent variables. For simplicity we also assume that there is no selection bias. Finally, we assume that the *Causal Markov Assumption* and the *Causal Faithfulness Assumption* [23] both hold. In other words, the conditional independences in the observational distribution correspond one-to-one with the d-separations in the causal DAG. Throughout the paper we represent variables with uppercase letters, while sets of variables are denoted by boldface. All proofs are provided in the Supplementary Material.

A directed edge $X \to Y$ in the causal DAG represents a *direct causal relation* between cause $X$ on effect $Y$. Intuitively, in this framework this indicates that manipulating $X$ will produce a change in $Y$, while manipulating $Y$ will have no effect on $X$. A more detailed discussion can be found in [23]. A sequence of directed edges $X_1 \to X_2 \to \cdots \to X_n$ is a *directed path*. If there exists a directed path from $X$ to $Y$ (or $X = Y$), then $X$ is an *ancestor* of $Y$ (denoted as $X \dashrightarrow Y$). Otherwise, $X$ is not an ancestor of $Y$ (denoted as $X \not\dashrightarrow Y$). For a set of variables $\boldsymbol{W}$, we write:

$$
\begin{aligned}
X \dashrightarrow \boldsymbol{W} &:= \exists Y \in \boldsymbol{W} : X \dashrightarrow Y, \\
X \not\dashrightarrow \boldsymbol{W} &:= \forall Y \in \boldsymbol{W} : X \not\dashrightarrow Y.
\end{aligned}
\tag{1}
$$

We define an *ancestral structure* as any non-strict partial order on the observed variables of the DAG, i.e., any relation that satisfies the following axioms:

$$(\textit{reflexivity}) : X \dashrightarrow X, \tag{2}$$

$$(\textit{transitivity}) : X \dashrightarrow Y \ \wedge \ Y \dashrightarrow Z \implies X \dashrightarrow Z, \tag{3}$$

$$(\textit{antisymmetry}) : X \dashrightarrow Y \wedge Y \dashrightarrow X \implies X = Y. \tag{4}$$

The underlying causal DAG induces a *unique* "true" ancestral structure, which represents the transitive closure of the direct causal relations projected on the observed variables.

For disjoint sets $X, Y, W$ we denote conditional independence of $X$ and $Y$ given $W$ as $X \perp\!\!\!\perp Y \mid W$, and conditional dependence as $X \not\perp\!\!\!\perp Y \mid W$. We call the cardinality $|W|$ the *order* of the conditional (in)dependence relation. Following [2] we define a *minimal conditional independence* by:

$$X \perp\!\!\!\perp Y \mid W \cup [Z] := (X \perp\!\!\!\perp Y \mid W \cup Z) \wedge (X \not\perp\!\!\!\perp Y \mid W),$$

and similarly, a *minimal conditional dependence* by:

$$X \not\perp\!\!\!\perp Y \mid W \cup [Z] := (X \not\perp\!\!\!\perp Y \mid W \cup Z) \wedge (X \perp\!\!\!\perp Y \mid W).$$

The square brackets indicate that $Z$ is needed for the (in)dependence to hold in the context of $W$. Note that the negation of a minimal conditional independence is not a minimal conditional dependence. Minimal conditional (in)dependences are closely related to ancestral relations, as pointed out in [2]:

**Lemma 1.** *For disjoint (sets of) variables $X, Y, Z, W$:*

$$X \perp\!\!\!\perp Y \mid W \cup [Z] \implies Z \dashrightarrow (\{X, Y\} \cup W), \tag{5}$$

$$X \not\perp\!\!\!\perp Y \mid W \cup [Z] \implies Z \not\dashrightarrow (\{X, Y\} \cup W). \tag{6}$$

Exploiting these rules (as well as others that will be introduced in Section 3) to deduce ancestral relations directly from (in)dependences is key to the greatly improved scalability of our method.

**Related work on conflict resolution**  One of the earliest algorithms to deal with conflicting inputs in constraint-based causal discovery is Conservative PC [18], which adds "redundant" checks to the PC algorithm that allow it to detect inconsistencies in the inputs, and then makes only predictions that do not rely on the ambiguous inputs. The same idea can be applied to FCI, yielding Conservative FCI (CFCI) [4, 10]. BCCD (Bayesian Constraint-based Causal Discovery) [3] uses Bayesian confidence estimates to process information in decreasing order of reliability, discarding contradictory inputs as they arise. COmbINE (Causal discovery from Overlapping INtErventions) [25] is an algorithm that combines the output of FCI on several overlapping observational and experimental datasets into a single causal model by first pooling and recalibrating the independence test $p$-values, and then adding each constraint incrementally in order of reliability to a SAT instance. Any constraint that makes the problem unsatisfiable is discarded.

Our approach is inspired by a method presented by Hyttinen, Eberhardt and Järvisalo [8] (that we will refer to as HEJ in this paper), in which causal discovery is formulated as a constrained discrete minimization problem. Given a list of weighted independence statements, HEJ searches for the optimal causal graph $\mathcal{G}$ (an acyclic directed mixed graph, or ADMG) that minimizes the sum of the weights of the independence statements that are violated according to $\mathcal{G}$. In order to test whether a causal graph $\mathcal{G}$ induces a certain independence, the method creates an *encoding DAG of d-connection graphs*. D-connection graphs are graphs that can be obtained from a causal graph through a series of operations (conditioning, marginalization and interventions). An encoding DAG of d-connection graphs is a complex structure encoding all possible d-connection graphs and the sequence of operations that generated them from a given causal graph. This approach has been shown to correct errors in the inputs, but is computationally demanding because of the huge search space.

## 3   ACI: Ancestral Causal Inference

We propose Ancestral Causal Inference (ACI), a causal discovery method that accurately reconstructs ancestral structures, also in the presence of latent variables and statistical errors. ACI builds on HEJ [8], but rather than optimizing over encoding DAGs, ACI optimizes over the much simpler (but still very expressive) ancestral structures.

For $n$ variables, the number of possible ancestral structures is the number of partial orders (`http://oeis.org/A001035`), which grows as $2^{n^2/4 + o(n^2)}$ [11], while the number of DAGs can be computed with a well-known super-exponential recurrence formula (`http://oeis.org/A003024`). The number of ADMGs is $|\mathrm{DAG}(n)| \times 2^{n(n-1)/2}$. Although still super-exponential, the number of ancestral structures grows asymptotically much slower than the number of DAGs and even more so, ADMGs. For example, for 7 variables, there are $6 \times 10^6$ ancestral structures but already $2.3 \times 10^{15}$ ADMGs, which lower bound the number of encoding DAGs of d-connection graphs used by HEJ.

**New rules** The rules in HEJ explicitly encode marginalization and conditioning operations on d-connection graphs, so they cannot be easily adapted to work directly with ancestral relations. Instead, ACI encodes the ancestral reasoning rules (2)–(6) and five novel causal reasoning rules:

**Lemma 2.** *For disjoint (sets) of variables* $X, Y, U, Z, \boldsymbol{W}$:

$$(X \perp\!\!\!\perp Y \mid \boldsymbol{Z}) \wedge (X \not\dashrightarrow \boldsymbol{Z}) \implies X \not\dashrightarrow Y, \tag{7}$$

$$X \not\perp\!\!\!\perp Y \mid \boldsymbol{W} \cup [Z] \implies X \not\perp\!\!\!\perp Z \mid \boldsymbol{W}, \tag{8}$$

$$X \perp\!\!\!\perp Y \mid \boldsymbol{W} \cup [Z] \implies X \not\perp\!\!\!\perp Z \mid \boldsymbol{W}, \tag{9}$$

$$(X \perp\!\!\!\perp Y \mid \boldsymbol{W} \cup [Z]) \wedge (X \perp\!\!\!\perp Z \mid \boldsymbol{W} \cup U) \implies (X \perp\!\!\!\perp Y \mid \boldsymbol{W} \cup U), \tag{10}$$

$$(Z \not\perp\!\!\!\perp X \mid \boldsymbol{W}) \wedge (Z \not\perp\!\!\!\perp Y \mid \boldsymbol{W}) \wedge (X \perp\!\!\!\perp Y \mid \boldsymbol{W}) \implies X \not\perp\!\!\!\perp Y \mid \boldsymbol{W} \cup Z. \tag{11}$$

We prove the soundness of the rules in the Supplementary Material. We elaborate some conjectures about their completeness in the discussion after Theorem 1 in the next Section.

**Optimization of loss function** We formulate causal discovery as an optimization problem where a loss function is optimized over possible causal structures. Intuitively, the loss function sums the weights of all the inputs that are violated in a candidate causal structure.

Given a list $I$ of weighted input statements $(i_j, w_j)$, where $i_j$ is the input statement and $w_j$ is the associated weight, we define the loss function as the sum of the weights of the input statements that are not satisfied in a given possible structure $W \in \mathcal{W}$, where $\mathcal{W}$ denotes the set of all possible causal structures. Causal discovery is formulated as a discrete optimization problem:

$$W^* = \arg\min_{W \in \mathcal{W}} \mathcal{L}(W; I), \tag{12}$$

$$\mathcal{L}(W; I) := \sum_{(i_j, w_j) \in I: \; W \cup \mathcal{R} \models \neg i_j} w_j, \tag{13}$$

where $W \cup \mathcal{R} \models \neg i_j$ means that input $i_j$ is not satisfied in structure $W$ according to the rules $\mathcal{R}$.

This general formulation includes both HEJ and ACI, which differ in the types of possible structures $\mathcal{W}$ and the rules $\mathcal{R}$. In HEJ $\mathcal{W}$ represents all possible causal graphs (specifically, acyclic directed mixed graphs, or ADMGs, in the acyclic case) and $\mathcal{R}$ are operations on d-connection graphs. In ACI $\mathcal{W}$ represent ancestral structures (defined with the rules(2)-(4)) and the rules $\mathcal{R}$ are rules (5)–(11).

**Constrained optimization in ASP** The constrained optimization problem in (12) can be implemented using a variety of methods. Given the complexity of the rules, a formulation in an expressive logical language that supports optimization, e.g., Answer Set Programming (ASP), is very convenient. ASP is a widely used declarative programming language based on the stable model semantics [12, 7] that has successfully been applied to several NP-hard problems. For ACI we use the state-of-the-art ASP solver `clingo 4` [6]. We provide the encoding in the Supplementary Material.

**Weighting schemes** ACI supports two types of input statements: conditional independences and ancestral relations. These statements can each be assigned a weight that reflects their confidence. We propose two simple approaches with the desirable properties of making ACI asymptotically consistent under mild assumptions (as described in the end of this Section), and assigning a much smaller weight to independences than to dependences (which agrees with the intuition that one is confident about a measured strong dependence, but not about independence vs. weak dependence). The approaches are:

- a *frequentist* approach, in which for any appropriate frequentist statistical test with independence as null hypothesis (resp. a non-ancestral relation), we define the weight:

$$w = |\log p - \log \alpha|, \text{ where } p = p\text{-value of the test}, \alpha = \text{significance level (e.g., 5\%)}; \tag{14}$$

- a *Bayesian* approach, in which the weight of each input statement $i$ using data set $\mathcal{D}$ is:

$$w = \log \frac{p(i|\mathcal{D})}{p(\neg i|\mathcal{D})} = \log \frac{p(\mathcal{D}|i)}{p(\mathcal{D}|\neg i)} \frac{p(i)}{p(\neg i)}, \tag{15}$$

where the prior probability $p(i)$ can be used as a tuning parameter.

Given observational and interventional data, in which each intervention has a single known target (in particular, it is not a *fat-hand* intervention [5]), a simple way to obtain a weighted ancestral statement $X \dashrightarrow Y$ is with a two-sample test that tests whether the distribution of $Y$ changes with respect to its observational distribution when intervening on $X$. This approach conveniently applies to various types of interventions: perfect interventions [16], soft interventions [14], mechanism changes [24], and activity interventions [15]. The two-sample test can also be implemented as an independence test that tests for the independence of $Y$ and $I_X$, the indicator variable that has value $0$ for observational samples and $1$ for samples from the interventional distribution in which $X$ has been intervened upon.

## 4  Scoring causal predictions

The constrained minimization in (12) may produce several optimal solutions, because the underlying structure may not be identifiable from the inputs. To address this issue, we propose to use the loss function (13) and score the confidence of a feature $f$ (e.g., an ancestral relation $X \dashrightarrow Y$) as:

$$C(f) = \min_{W \in \mathcal{W}} \mathcal{L}(W; I \cup \{(\neg f, \infty)\}) - \min_{W \in \mathcal{W}} \mathcal{L}(W; I \cup \{(f, \infty)\}). \tag{16}$$

Without going into details here, we note that the confidence (16) can be interpreted as a MAP approximation of the log-odds ratio of the probability that feature $f$ is true in a Markov Logic model:

$$\frac{\mathbb{P}(f \mid I, \mathcal{R})}{\mathbb{P}(\neg f \mid I, \mathcal{R})} = \frac{\sum_{W \in \mathcal{W}} e^{-\mathcal{L}(W; I)} 1_{W \cup \mathcal{R} \models f}}{\sum_{W \in \mathcal{W}} e^{-\mathcal{L}(W; I)} 1_{W \cup \mathcal{R} \models \neg f}} \approx \frac{\max_{W \in \mathcal{W}} e^{-\mathcal{L}(W; I \cup \{(f, \infty)\})}}{\max_{W \in \mathcal{W}} e^{-\mathcal{L}(W; I \cup \{(\neg f, \infty)\})}} = e^{C(f)}.$$

In this paper, we usually consider the features $f$ to be ancestral relations, but the idea is more generally applicable. For example, combined with HEJ it can be used to score direct causal relations.

**Soundness and completeness**    Our scoring method is sound for oracle inputs:

**Theorem 1.** *Let $\mathcal{R}$ be sound (not necessarily complete) causal reasoning rules. For any feature $f$, the confidence score $C(f)$ of (16) is sound for oracle inputs with infinite weights.*

Here, soundness means that $C(f) = \infty$ if $f$ is identifiable from the inputs, $C(f) = -\infty$ if $\neg f$ is identifiable from the inputs, and $C(f) = 0$ otherwise (neither are identifiable). As features, we can consider for example ancestral relations $f = X \dashrightarrow Y$ for variables $X, Y$. We conjecture that the rules (2)–(11) are "order-1-complete", i.e., they allow one to deduce all (non)ancestral relations that are identifiable from oracle conditional independences of order $\leq 1$ in observational data. For higher-order inputs additional rules can be derived. However, our primary interest in this work is improving computation time and accuracy, and we are willing to sacrifice completeness. A more detailed study of the completeness properties is left as future work.

**Asymptotic consistency**    Denote the number of samples by $N$. For the frequentist weights in (14), we assume that the statistical tests are consistent in the following sense:

$$\log p_N - \log \alpha_N \xrightarrow{P} \begin{cases} -\infty & H_1 \\ +\infty & H_0, \end{cases} \tag{17}$$

as $N \to \infty$, where the null hypothesis $H_0$ is independence/nonancestral relation and the alternative hypothesis $H_1$ is dependence/ancestral relation. Note that we need to choose a sample-size dependent threshold $\alpha_N$ such that $\alpha_N \to 0$ at a suitable rate. Kalisch and Bühlmann [9] show how this can be done for partial correlation tests under the assumption that the distribution is multivariate Gaussian.

For the Bayesian weighting scheme in (15), we assume that for $N \to \infty$,

$$w_N \xrightarrow{P} \begin{cases} -\infty & \text{if } i \text{ is true} \\ +\infty & \text{if } i \text{ is false.} \end{cases} \tag{18}$$

This will hold (as long as there is no model misspecification) under mild technical conditions for finite-dimensional exponential family models. In both cases, the probability of a type I or type II error will converge to 0, and in addition, the corresponding weight will converge to $\infty$.

**Theorem 2.** *Let $\mathcal{R}$ be sound (not necessarily complete) causal reasoning rules. For any feature $f$, the confidence score $C(f)$ of (16) is asymptotically consistent under assumption (17) or (18).*

Here, "asymptotically consistent" means that the confidence score $C(f) \to \infty$ in probability if $f$ is identifiably true, $C(f) \to -\infty$ in probability if $f$ is identifiably false, and $C(f) \to 0$ in probability otherwise.

| Average execution time (s) | | | | | |
|---|---|---|---|---|---|
| $n$ | $c$ | ACI | HEJ | BAFCI | BACFCI |
| 6 | 1 | 0.21 | 12.09 | 8.39 | 12.51 |
| 6 | 4 | 1.66 | 432.67 | 11.10 | 16.36 |
| 7 | 1 | 1.03 | 715.74 | 9.37 | 15.12 |
| 8 | 1 | 9.74 | $\geq 2500$ | 13.71 | 21.71 |
| 9 | 1 | 146.66 | $\gg 2500$ | 18.28 | 28.51 |

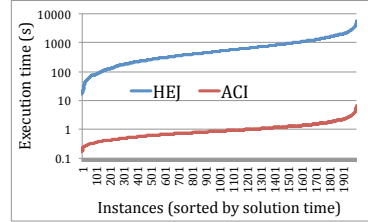

(a)                                        (b)

Figure 1: Execution time comparison on synthetic data for the frequentist test on 2000 synthetic models: (a) average execution time for different combinations of number of variables $n$ and max. order $c$; (b) detailed plot of execution times for $n = 7, c = 1$ (logarithmic scale).

## 5 Evaluation

In this section we report evaluations on synthetically generated data and an application on a real dataset. Crucially, in causal discovery precision is often more important than recall. In many real-world applications, discovering a few high-confidence causal relations is more useful than finding every possible causal relation, as reflected in recently proposed algorithms, e.g., [17].

**Compared methods** We compare the predictions of ACI and of the acyclic causally insufficient version of HEJ [8], when used in combination with our scoring method (16). We also evaluate two standard methods: Anytime FCI [22, 26] and Anytime CFCI [4], as implemented in the `pcalg` R package [10]. We use the anytime versions of (C)FCI because they allow for independence test results up to a certain order. We obtain the ancestral relations from the output PAG using Theorem 3.1 from [20]. (Anytime) FCI and CFCI do not rank their predictions, but only predict the type of relation: ancestral (which we convert to +1), non-ancestral (-1) and unknown (0). To get a scoring of the predictions, we also compare with bootstrapped versions of Anytime FCI and Anytime CFCI. We perform the bootstrap by repeating the following procedure 100 times: sample randomly half of the data, perform the independence tests, run Anytime (C)FCI. From the 100 output PAGs we extract the ancestral predictions and average them. We refer to these methods as BA(C)FCI. For a fair comparison, we use the same independence tests and thresholds for all methods.

**Synthetic data** We simulate the data using the simulator from HEJ [8]: for each experimental condition (e.g., a given number of variables $n$ and order $c$), we generate randomly $M$ linear acyclic models with latent variables and Gaussian noise and sample $N = 500$ data points. We then perform independence tests up to order $c$ and weight the (in)dependence statements using the weighting schemes described in Section 3. For the frequentist weights we use tests based on partial correlations and Fisher's $z$-transform to obtain approximate $p$-values (see, e.g., [9]) with significance level $\alpha = 0.05$. For the Bayesian weights, we use the Bayesian test for conditional independence presented in [13] as implemented by HEJ with a prior probability of 0.1 for independence.

In Figure 1(a) we show the average execution times on a single core of a 2.80GHz CPU for different combinations of $n$ and $c$, while in Figure 1(b) we show the execution times for $n = 7, c = 1$, sorting the execution times in ascending order. For 7 variables ACI is almost 3 orders of magnitude faster than HEJ, and the difference grows exponentially as $n$ increases. For 8 variables HEJ can complete only four of the first 40 simulated models before the timeout of 2500s. For reference we add the execution time for bootstrapped anytime FCI and CFCI.

In Figure 2 we show the accuracy of the predictions with precision-recall (PR) curves for both ancestral ($X \dashrightarrow Y$) and nonancestral ($X \not\dashrightarrow Y$) relations, in different settings. In this Figure, for ACI and HEJ all of the results are computed using frequentist weights and, as in all evaluations, our scoring method (16). While for these two methods we use $c = 1$, for (bootstrapped) (C)FCI we use all possible independence test results ($c = n - 2$). In this case, the anytime versions of FCI and CFCI are equivalent to the standard versions of FCI and CFCI. Since the overall results are similar, we report the results with the Bayesian weights in the Supplementary Material.

In the first row of Figure 2, we show the setting with $n = 6$ variables. The performances of HEJ and ACI coincide, performing significantly better for nonancestral predictions and the top ancestral

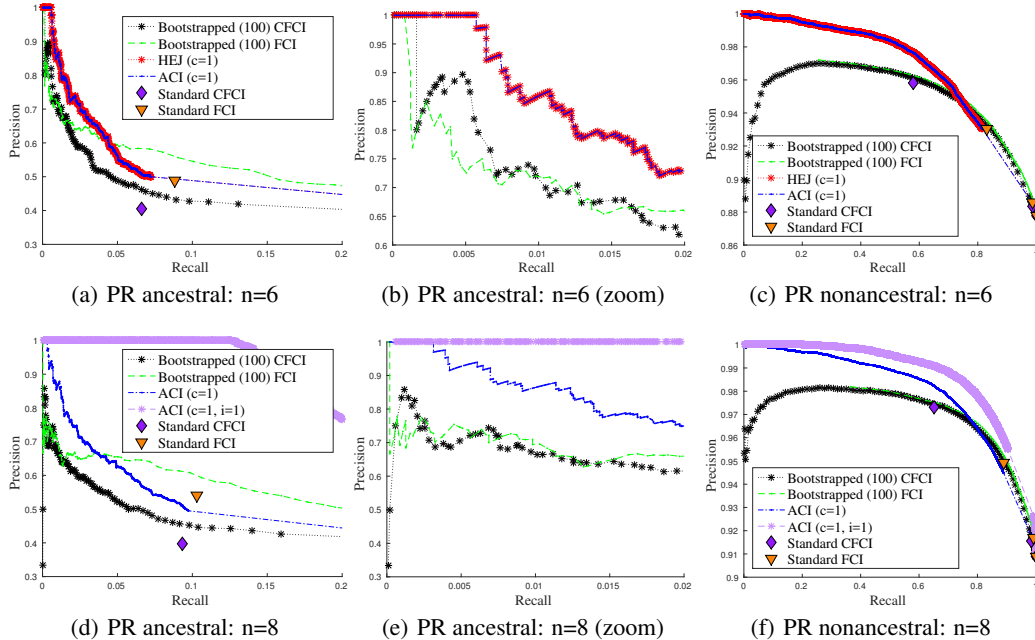

(a) PR ancestral: n=6     (b) PR ancestral: n=6 (zoom)     (c) PR nonancestral: n=6

(d) PR ancestral: n=8     (e) PR ancestral: n=8 (zoom)     (f) PR nonancestral: n=8

Figure 2: Accuracy on synthetic data for the two prediction tasks (ancestral and nonancestral relations) using the frequentist test with $\alpha = 0.05$. The left column shows the precision-recall curve for ancestral predictions, the middle column shows a zoomed-in version in the interval (0,0.02), while the right column shows the nonancestral predictions.

predictions (see zoomed-in version in Figure 2(b)). This is remarkable, as HEJ and ACI use only independence test results up to order $c = 1$, in contrast with (C)FCI which uses independence test results of all orders. Interestingly, the two discrete optimization algorithms do not seem to benefit much from higher order independence tests, thus we omit them from the plots (although we add the graphs in the Supplementary Material). Instead, bootstrapping traditional methods, oblivious to the (in)dependence weights, seems to produce surprisingly good results. Nevertheless, both ACI and HEJ outperform bootstrapped FCI and CFCI, suggesting these methods achieve nontrivial error-correction.

In the second row of Figure 2, we show the setting with 8 variables. In this setting HEJ is too slow. In addition to the previous plot, we plot the accuracy of ACI when there is oracle background knowledge on the descendants of one variable ($i = 1$). This setting simulates the effect of using interventional data, and we can see that the performance of ACI improves significantly, especially in the ancestral preditions. The performance of (bootstrapped) FCI and CFCI is limited by the fact that they cannot take advantage of this background knowledge, except with complicated postprocessing [1].

**Application on real data** We consider the challenging task of reconstructing a signalling network from flow cytometry data [21] under different experimental conditions. Here we consider one experimental condition as the observational setting and seven others as interventional settings. More details and more evaluations are reported in the Supplementary Material. In contrast to likelihood-based approaches like [21, 5, 15, 19], in our approach we do not need to model the interventions quantitatively. We only need to know the intervention *targets*, while the intervention *types* do not matter. Another advantage of our approach is that it takes into account possible latent variables.

We use a $t$-test to test for each intervention and for each variable whether its distribution changes with respect to the observational condition. We use the $p$-values of these tests as in (14) in order to obtain weighted ancestral relations that are used as input (with threshold $\alpha = 0.05$). For example, if adding U0126 (a MEK inhibitor) changes the distribution of RAF significantly with respect to the observational baseline, we get a weighted ancestral relation MEK--→RAF. In addition, we use partial correlations up to order 1 (tested in the observational data only) to obtain weighted independences used as input. We use ACI with (16) to score the ancestral relations for each ordered pair of variables. The main results are illustrated in Figure 3, where we compare ACI with bootstrapped anytime CFCI

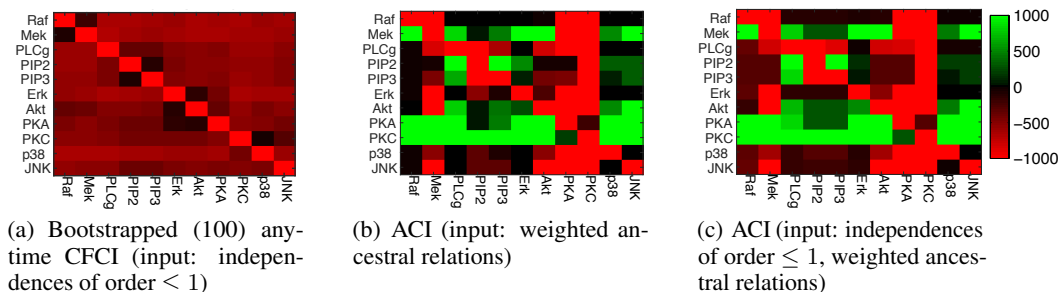

(a) Bootstrapped (100) any-time CFCI (input: independences of order ≤ 1)

(b) ACI (input: weighted ancestral relations)

(c) ACI (input: independences of order ≤ 1, weighted ancestral relations)

Figure 3: Results for flow cytometry dataset. Each matrix represents the ancestral relations, where each row represents a cause and each column an effect. The colors encode the confidence levels: green is positive, black is unknown, while red is negative. The intensity of the color represents the degree of confidence. For example, ACI identifies MEK to be a cause of RAF with high confidence.

under different inputs. The output for boostrapped anytime FCI is similar, so we report it only in the Supplementary Material. Algorithms like (anytime) (C)FCI can only use the independences in the observational data as input and therefore miss the strongest signal, *weighted ancestral relations*, which are obtained by comparing interventional with observational data. In the Supplementary Material, we compare also with other methods ([17], [15]). Interestingly, as we show there, our results are similar to the best acyclic model reconstructed by the score-based method from [15]. As for other constraint-based methods, HEJ is computationally unfeasible in this setting, while COMBINE assumes perfect interventions (while this dataset contains mostly activity interventions).

Notably, our algorithms can correctly recover from faithfulness violations (e.g., the independence between MEK and ERK), because they take into account the weight of the input statements (the weight of the independence is considerably smaller than that of the ancestral relation, which corresponds with a quite significant change in distribution). In contrast, methods that start by reconstructing the skeleton, like (anytime) (C)FCI, would decide that MEK and ERK are nonadjacent, and are unable to recover from that erroneous decision. This illustrates another advantage of our approach.

## 6   Discussion and conclusions

As we have shown, ancestral structures are very well-suited for causal discovery. They offer a natural way to incorporate background causal knowledge, e.g., from experimental data, and allow a huge computational advantage over existing representations for error-correcting algorithms, such as [8]. When needed, ancestral structures can be mapped to a finer-grained representation with direct causal relations, as we sketch in the Supplementary Material. Furthermore, confidence estimates on causal predictions are extremely helpful in practice, and can significantly boost the reliability of the output. Although standard methods, like bootstrapping (C)FCI, already provide reasonable estimates, methods that take into account the confidence in the inputs, as the one presented here, can lead to further improvements of the reliability of causal relations inferred from data.

Strangely (or fortunately) enough, neither of the optimization methods seems to improve much with higher order independence test results. We conjecture that this may happen because our loss function essentially assumes that the test results are independent from another (which is not true). Finding a way to take this into account in the loss function may further improve the achievable accuracy, but such an extension may not be straightforward.

### Acknowledgments

SM and JMM were supported by NWO, the Netherlands Organization for Scientific Research (VIDI grant 639.072.410). SM was also supported by the Dutch programme COMMIT/ under the Data2Semantics project. TC was supported by NWO grant 612.001.202 (MoCoCaDi), and EU-FP7 grant agreement n.603016 (MATRICS). We also thank Sofia Triantafillou for her feedback, especially for pointing out the correct way to read ancestral relations from a PAG.

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
