[Supplementary Material]

# Supplementary Material for "Ancestral Causal Inference", NIPS 2016

**Sara Magliacane**
VU University Amsterdam, University of Amsterdam
`sara.magliacane@gmail.com`

**Tom Claassen**
Radboud University Nijmegen
`tomc@cs.ru.nl`

**Joris M. Mooij**
University of Amsterdam
`j.m.mooij@uva.nl`

## 1 Proofs

### 1.1 ACI causal reasoning rules

We give a combined proof of all the ACI reasoning rules. Note that the numbering of the rules here is different from the numbering used in the main paper.

**Lemma 1.** *For $X$, $Y$, $Z$, $U$, $\boldsymbol{W}$ disjoint (sets of) variables:*

*1.* $(X \perp\!\!\!\perp Y \mid \boldsymbol{W}) \wedge (X \not\dashrightarrow \boldsymbol{W}) \implies X \not\dashrightarrow Y$

*2.* $X \not\perp\!\!\!\perp Y \mid \boldsymbol{W} \cup [Z] \implies (X \not\perp\!\!\!\perp Z \mid \boldsymbol{W}) \wedge (Z \not\dashrightarrow \{X,Y\} \cup \boldsymbol{W})$

*3.* $X \perp\!\!\!\perp Y \mid \boldsymbol{W} \cup [Z] \implies (X \not\perp\!\!\!\perp Z \mid \boldsymbol{W}) \wedge (Z \dashrightarrow \{X,Y\} \cup \boldsymbol{W})$

*4.* $(X \perp\!\!\!\perp Y \mid \boldsymbol{W} \cup [Z]) \wedge (X \perp\!\!\!\perp Z \mid \boldsymbol{W} \cup U) \implies (X \perp\!\!\!\perp Y \mid \boldsymbol{W} \cup U)$

*5.* $(Z \not\perp\!\!\!\perp X \mid \boldsymbol{W}) \wedge (Z \not\perp\!\!\!\perp Y \mid \boldsymbol{W}) \wedge (X \perp\!\!\!\perp Y \mid \boldsymbol{W}) \implies X \not\perp\!\!\!\perp Y \mid \boldsymbol{W} \cup Z$

*Proof.* We assume a causal DAG with possible latent variables, the causal Markov assumption, and the causal faithfulness assumption.

1. This is a strengthened version of rule $\mathcal{R}2$(i) in [3]: note that the additional assumptions made there ($Y \not\dashrightarrow \boldsymbol{W}$, $Y \not\dashrightarrow X$) are redundant and not actually used in their proof. For completeness, we give the proof here. If $X \dashrightarrow Y$, then there is a directed path from $X$ to $Y$. As all paths between $X$ and $Y$ are blocked by $\boldsymbol{W}$, the directed path from $X$ to $Y$ must contain a node $W \in \boldsymbol{W}$. Hence $X \dashrightarrow W$, a contradiction with $X \not\dashrightarrow \boldsymbol{W}$.

2. If $X \not\perp\!\!\!\perp Y \mid \boldsymbol{W} \cup [Z]$ then there exists a path $\pi$ between $X$ and $Y$ such that each noncollider on $\pi$ is not in $\boldsymbol{W} \cup \{Z\}$, every collider on $\pi$ is ancestor of $\boldsymbol{W} \cup \{Z\}$, and there exists a collider on $\pi$ that is ancestor of $Z$ but not of $\boldsymbol{W}$. Let $C$ be the collider on $\pi$ closest to $X$ that is ancestor of $Z$ but not of $\boldsymbol{W}$. Note that

    (a) The path $X \cdots C \rightarrow \cdots \rightarrow Z$ is d-connected given $\boldsymbol{W}$.
    (b) $Z \not\dashrightarrow \boldsymbol{W}$ (because otherwise $C \dashrightarrow Z \dashrightarrow \boldsymbol{W}$, a contradiction).
    (c) $Z \not\dashrightarrow Y$ (because otherwise the path $X \cdots C \rightarrow \cdots \rightarrow Z \rightarrow \cdots \rightarrow Y$ would be d-connected given $\boldsymbol{W}$, a contradiction).

    Hence we conclude that $X \not\perp\!\!\!\perp Z \mid \boldsymbol{W}$, $Z \not\dashrightarrow \boldsymbol{W}$, $Z \not\dashrightarrow Y$, and by symmetry also $Z \not\dashrightarrow X$.

3. Suppose $X \perp\!\!\!\perp Y \mid \boldsymbol{W} \cup [Z]$. Then there exists a path $\pi$ between $X$ and $Y$, such that each noncollider on $\pi$ is not in $\boldsymbol{W}$, each collider on $\pi$ is an ancestor of $\boldsymbol{W}$, and $Z$ is a noncollider on $\pi$. Note that

   (a) The subpath $X \cdots Z$ must be d-connected given $\boldsymbol{W}$.

   (b) $Z$ has at least one outgoing edge on $\pi$. Follow this edge further along $\pi$ until reaching either $X, Y$, or the first collider. When a collider is reached, follow the directed path to $\boldsymbol{W}$. Hence there is a directed path from $Z$ to $X$ or $Y$ or to $\boldsymbol{W}$, i.e., $Z \dashrightarrow \{X, Y\} \cup \boldsymbol{W}$.

4. If in addition, $X \perp\!\!\!\perp Z \mid \boldsymbol{W} \cup U$, then $U$ must be a noncollider on the subpath $X \cdots Z$. Therefore, $X \perp\!\!\!\perp Y \mid \boldsymbol{W} \cup U$.

5. Assume that $Z \not\perp\!\!\!\perp X \mid \boldsymbol{W}$ and $Z \not\perp\!\!\!\perp Y \mid \boldsymbol{W}$. Then there must be paths $\pi$ between $Z$ and $X$ and $\rho$ between $Z$ and $Y$ such that each noncollider is not in $\boldsymbol{W}$ and each collider is ancestor of $\boldsymbol{W}$. Let $U$ be the node on $\pi$ closest to $X$ that is also on $\rho$ (this could be $Z$). Then we have a path $X \cdots U \cdots Y$ such that each collider (except $U$) is ancestor of $\boldsymbol{W}$ and each noncollider (except $U$) is not in $\boldsymbol{W}$. This path must be blocked given $\boldsymbol{W}$ as $X \perp\!\!\!\perp Y \mid \boldsymbol{W}$. If $U$ would be a noncollider on this path, it would need to be in $\boldsymbol{W}$ in order to block it; however, it must then also be a noncollider on $\pi$ or $\rho$ and hence cannot be in $\boldsymbol{W}$. Therefore, $U$ must be a collider on this path and cannot be ancestor of $\boldsymbol{W}$. We have to show that $U$ is ancestor of $Z$. If $U$ were a collider on $\pi$ or $\rho$, it would be ancestor of $\boldsymbol{W}$, a contradiction. Hence $U$ must have an outgoing arrow pointing towards $Z$ on $\pi$ and $\rho$. If we encounter a collider following the directed edges, we get a contradiction, as that collider, and hence $U$, would be ancestor of $\boldsymbol{W}$. Hence $U$ is ancestor of $Z$, and therefore, $X \not\perp\!\!\!\perp Y \mid \boldsymbol{W} \cup Z$.

$\square$

## 1.2 Soundness

**Theorem 1.** *Let $\mathcal{R}$ be sound (not necessarily complete) causal reasoning rules. For any feature $f$, the confidence score $C(f)$ of (16) is sound for oracle inputs with infinite weights, i.e., $C(f) = \infty$ if $f$ is identifiable from the inputs, $C(f) = -\infty$ if $\neg f$ is identifiable from the inputs, and $C(f) = 0$ otherwise (neither are identifiable).*

*Proof.* We assume that the data generating process is described by a causal DAG which may contain additional latent variables, and that the distributions are faithful to the DAG. The theorem then follows directly from the soundness of the rules and the soundness of logical reasoning. $\square$

## 1.3 Asymptotic consistency of scoring method

**Theorem 2.** *Let $\mathcal{R}$ be sound (not necessarily complete) causal reasoning rules. For any feature $f$, the confidence score $C(f)$ of (16) is asymptotically consistent under assumption (14) or (15) in the main paper, i.e.,*

- *$C(f) \to \infty$ in probability if $f$ is identifiably true,*

- *$C(f) \to -\infty$ in probability if $f$ is identifiably false,*

- *$C(f) \to 0$ in probability otherwise (neither are identifiable).*

*Proof.* As the number of statistical tests is fixed (or at least bounded from above), the probability of *any* error in the test results converges to 0 asymptotically. The loss function of all structures that do not correspond with the properties of the true causal DAG converges to $+\infty$ in probability, whereas the loss function of all structures that are compatible with properties of the true causal DAG converges to 0 in probability. $\square$

(a) PR ancestral

(b) PR ancestral (zoom)

(c) PR nonancestral

Figure 1: Synthetic data: accuracy for the two prediction tasks (ancestral and nonancestral relations) for $n = 6$ variables using the frequentist test with $\alpha = 0.05$, also for higher order $c$.

## 2 Additional results on synthetic data

In Figures 1 and 2 we show the performance of ACI and HEJ [6] for higher order independence test results ($c = 4$). As in the main paper, for (bootstrapped) FCI and CFCI we use $c = 4$, because it gives the best predictions for these methods. In Figure 1 we report more accuracy results on the frequentist test with $\alpha = 0.05$, the same setting as Figure 2 (a-c) in the main paper. As we see, the performances of ACI and HEJ do not really improve with higher order but actually seem to deteriorate.

(a) PR ancestral

(b) PR ancestral (zoom)

(c) PR nonancestral

Figure 2: Synthetic data: accuracy for the two prediction tasks (ancestral and nonancestral relations) for $n = 6$ variables using the Bayesian test with prior probability of independence $p = 0.1$.

In Figure 2 we report accuracy results on synthetic data also for the Bayesian test described in the main paper, with prior probability of independence $p = 0.1$. Using the Bayesian test does not change the overall conclusions: ACI and HEJ overlap for order $c = 1$ and they perform better than bootstrapped (C)FCI.

Table 1: Reagents used in the various experimental conditions in [11] and corresponding intervention types and targets. The intervention types and targets are based on (our interpretation of) biological background knowledge. The upper table describes the "no-ICAM" batch of conditions that is most commonly used in the literature. The lower table describes the additional "ICAM" batch of conditions that we also use here.

|  | Reagents | | | Intervention | |
|---|---|---|---|---|---|
|  | $\alpha$-CD3, $\alpha$-CD28 | ICAM-2 | Additional | Target | Type |
| no-ICAM: | + | - | - | - | (observational) |
|  | + | - | AKT inhibitor | AKT | activity |
|  | + | - | G0076 | PKC | activity |
|  | + | - | Psitectorigenin | PIP2 | abundance |
|  | + | - | U0126 | MEK | activity |
|  | + | - | LY294002 | PIP2/PIP3 | mechanism change |
|  | - | - | PMA | PKC | activity + fat-hand |
|  | - | - | $\beta$2CAMP | PKA | activity + fat-hand |

|  | Reagents | | | Intervention | |
|---|---|---|---|---|---|
|  | $\alpha$-CD3, $\alpha$-CD28 | ICAM-2 | Additional | Target | Type |
| ICAM: | + | + | - | - | (observational) |
|  | + | + | AKT inhibitor | AKT | activity |
|  | + | + | G0076 | PKC | activity |
|  | + | + | Psitectorigenin | PIP2 | abundance |
|  | + | + | U0126 | MEK | activity |
|  | + | + | LY294002 | PIP2/PIP3 | mechanism change |
|  | - | - | PMA | PKC | activity + fat-hand |
|  | - | - | $\beta$2CAMP | PKA | activity + fat-hand |

# 3   Application on real data

We provide more details and more results on the real-world dataset that was briefly described in the main paper, the flow cytometry data [11]. The data consists of simultaneous measurements of expression levels of 11 biochemical agents in individual cells of the human immune system under 14 different experimental conditions.

## 3.1   Experimental conditions

The experimental conditions can be grouped into two batches of 8 conditions each that have very similar interventions:

- "no-ICAM", used in the main paper and commonly used in the literature;
- "ICAM", where Intercellular Adhesion Protein-2 (ICAM-2) was added (except when PMA or $\beta$2CAMP was added).

For each batch of 8 conditions, the experimenters added $\alpha$-CD3 and $\alpha$-CD28 to activate the signaling network in 6 out of 8 conditions. For the remaining two conditions (PMA and $\beta$2CAMP), $\alpha$-CD3 and $\alpha$-CD28 were not added (and neither was ICAM-2). We can consider the *absence* of these stimuli as a global intervention relative to the observational baseline (where $\alpha$-CD3 and $\alpha$-CD28 are present, and in addition ICAM-2 is present in the ICAM batch). For each batch (ICAM and no-ICAM), we can consider an observational dataset and 7 interventional datasets with different activators and inhibitors added to the cells, as described in Table 1. Note that the datasets from the last two conditions are the same in both settings. For more information about intervention types, see [9].

In this paper, we ignore the fact that in the last two interventional datasets in each batch (PMA and $\beta$2CAMP) there is also a global intervention. Ignoring the global intervention allows us to compute the weighted ancestral relations, since we consider any variable that changes its distribution with respect to the observational condition to be an effect of the main target of the intervention (PKC for PMA and PKA for $\beta$2CAMP). This is in line with previous work [11, 9]. Also, we consider only

(a) Independences of order 0

(b) Weighted ancestral relations

(d) ACI (input: independences or-
der ≤ 1)

(e) ACI (input: weighted ances-
tral relations)

(f) ACI (input: independences or-
der ≤ 1, weighted ancestral rela-
tions)

(g) Bootstrapped (100) anytime
FCI (input: independences order
≤ 1)

(h) Bootstrapped (100) anytime
CFCI (input: independences or-
der ≤ 1)

Figure 3: Results on flow cytometry dataset, no-ICAM batch. The top row represents some of the possible inputs: weighted independences of order 0 from the observational dataset (the inputs include also order 1 test results, but these are not visualized here) and weighted ancestral relations recovered from comparing the interventional datasets with the observational data. In the bottom two rows each matrix represents the ancestral relations that are estimated using different inputs and different methods (ACI, bootstrapped anytime FCI or CFCI). Each row represents a cause, while the columns are the effects. The colors encodes the confidence levels, green is positive, black is unknown, while red is negative. The intensity of the color represents the degree of confidence.

PIP3 as the main target of the LY294002 intervention, based on the consensus network [11], even though in [9] both PIP2 and PIP3 are considered to be targets of this intervention. In future work, we plan to extend ACI in order to address the task of learning the intervention targets from data, as done by [2] for a score-based approach.

In the main paper we provide some results for the most commonly used no-ICAM batch of experimental conditions. Below we report additional results on the same batch. Moreover, we provide results for causal discovery on the ICAM batch, which are quite consistent with the no-ICAM batch. Finally, we compare with other methods that were applied to this dataset, especially with a score-based approach ([9]) that shows surprisingly similar results to ACI, although it uses a very different method.

## 3.2 Results on no-ICAM batch

In Figure 3 we provide additional results for the no-ICAM batch. In the first row we show some of the possible inputs: weighted independences (in this case partial correlations) from observational data and weighted ancestral relations from comparing the interventional datasets with the observational

(a) Independences of order 0     (b) Weighted ancestral relations

(d) ACI (input: independences order $\leq 1$)

(e) ACI (input: weighted ancestral relations)

(f) ACI (input: independences order $\leq 1$, weighted ancestral relations)

(g) Bootstrapped (100) anytime FCI(input: independences order $\leq 1$)

(h) Bootstrapped (100) anytime CFCI (input: independences order $\leq 1$)

Figure 4: Results on flow cytometry dataset, ICAM batch. Same comparison as in Figure 3, but for the ICAM batch.

data. Specifically, we consider as inputs only independences up to order 1 (but only independences of order 0 are visualized in the figure). The color encodes the weight of the independence. As an example, the heatmap shows that Raf and Mek are strongly dependent.

For the weighted ancestral relations, in Figure 3 we plot a matrix in which each row represents a cause, while the columns are the effects. As described in the main paper we use a $t$-test to test for each intervention and for each variable whether its distribution changes with respect to the observational condition. We use the biological knowledge summarised in Table 1 to define the intervention target, which is then considered the putative "cause". Then we use the $p$-values of these tests and a threshold $\alpha = 0.05$ to obtain the weights of the ancestral relations, similarly to what is proposed in the main paper for the frequentist weights for the independence tests:

$$w = |\log p - \log \alpha|.$$

For example, if adding U0126 (which is known to be a MEK inhibitor) changes the distribution of RAF with $p = 0.01$ with respect to the observational baseline, we get a weighted ancestral relation (MEK--→RAF, 1.609).

### 3.3 ICAM batch

In Figure 4 we show the results for the ICAM setting. These results are very similar to the results for the no-ICAM batch (see also Figure 5), showing that the predicted ancestral relations are robust. In particular it is clear that also for the ICAM batch, weighted ancestral relations are a very strong

Figure 5: ACI results (input: independences of order $\leq 1$ and weighted ancestral relations) on no-ICAM (left) and ICAM (right) batches. These heatmaps are identical to the ones in Figures 3 and 4, but are reproduced here next to each other for easy comparison.

signal, and that methods that can exploit them (e.g., ACI) have a distinct advantage over methods that cannot (e.g., FCI and CFCI).

In general, in both settings there appear to be various faithfulness violations. For example, it is well-known that MEK causes ERK, yet in the observational data these two variables are independent. Nevertheless, we can see in the data that an intervention on MEK leads to a change of ERK, as expected. It is interesting to note that our approach can correctly recover from this faithfulness violation because it takes into account the weight of the input statements (note that the weight of the independence is smaller than that of the ancestral relation, which corresponds with a quite significant change in distribution). In contrast, methods that start by reconstructing the skeleton (like (C)FCI or LoCI [1]) would decide that MEK and ERK are nonadjacent, unable to recover from that erroneous decision. This illustrates one of the advantages of our approach.

### 3.4 Comparison with other approaches

We also compare our results with other, mostly score-based approaches. Amongst other results, [9] report the top 17 direct causal relations on the no-ICAM batch that were inferred by their score-based method when assuming acyclicity. In order to compare fairly with the ancestral relations found by ACI, we first perform a transitive closure of these direct causal relations, which results in 21 ancestral relations. We then take the top 21 predicted ancestral relations from ACI (for the same no-ICAM batch), and compare the two in Figure 6. The black edges, the majority, represent the ancestral relations found by both methods. The blue edges are found only by ACI, while the grey edges are found only by [9]. Interestingly, the results are quite similar, despite the very different approaches. In particular, ACI allows for confounders and is constraint-based, while the method in [9] assumes causal sufficiency (i.e., no confounders) and is score-based.

Table 2 summarizes most of the existing work on this flow cytometry dataset. It was originally part of the S1 material of [8]. We have updated it here by adding also the results for ACI and the transitive closure of [9].

## 4 Mapping ancestral structures to direct causal relations

An ancestral structure can be seen as the transitive closure of the directed edges of an acyclic directed mixed graph (ADMG). There are several strategies to reconstruct "direct" causal relations from an ancestral structure, in particular in combination with our scoring method. Here we sketch a possible strategy, but we leave a more in-depth investigation to future work.

A possible strategy is to first recover the ancestral structure from ACI with our scoring method and then use it as "oracle" input constraints for the HEJ [6] algorithm. Specifically, for each weighted output $(X \dashrightarrow Y, w)$ obtained by ACI, we add $(X \dashrightarrow Y, \infty)$ to the input list $I$, and similarly for each $X \not\dashrightarrow Y$. Then we can use our scoring algorithm with HEJ to score direct causal relations (e.g., $f = X \to Y$) and direct acausal relations (e.g., $f = X \not\to Y$):

$$C(f) = \min_{W \in \mathcal{W}} \mathrm{loss}(W; I \cup \{(\neg f, \infty)\}) - \min_{W \in \mathcal{W}} \mathrm{loss}(W; I \cup \{(f, \infty)\}). \tag{1}$$

In the standard HEJ algorithm, $\mathcal{W}$ are all possible ADMGs, but with our additional constraints we can reduce the search space to only the ones that fit the specific ancestral structure, which is on average

Table 2: Updated Table S1 from [8]: causal relationships between the biochemical agents in the flow cytometry data of [11], according to different causal discovery methods. The consensus network according to [11] is denoted here by "[11]a" and their reconstructed network by "[11]b". For [9] we provide two versions: "[9]a" for the top 17 edges in the acyclic case, as reported in the original paper, and "[9]b" for its transitive closure, which consists of 21 edges. To provide a fair comparison, we also pick the top 21 ancestral predictions from ACI.

| Edge | Direct causal predictions | | | | | Ancestral predictions | | |
|---|---|---|---|---|---|---|---|---|
| | [11]a | [11]b | [9]a | [2] | ICP [10] | hiddenICP [10] | [9]b | ACI (top 21) |
| RAF→MEK | ✓ | ✓ | | | | ✓ | | |
| MEK→RAF | | | ✓ | ✓ | | ✓ | ✓ | ✓ |
| MEK→ERK | ✓ | ✓ | ✓ | | | | ✓ | ✓ |
| MEK→AKT | | | | | | | | ✓ |
| MEK→JNK | | | | | | | | ✓ |
| PLCg→PIP2 | ✓ | ✓ | | ✓ | ✓ | ✓ | | |
| PLCg→PIP3 | | ✓ | | ✓ | | | | |
| PLCg→PKC | ✓ | | | ✓ | | | | |
| PIP2→PLCg | | | ✓ | | ✓ | | ✓ | ✓ |
| PIP2→PIP3 | | | | ✓ | | | | |
| PIP2→PKC | ✓ | | | | | | | |
| PIP3→PLCg | ✓ | | | | | | ✓ | |
| PIP3→PIP2 | ✓ | ✓ | ✓ | | ✓ | ✓ | ✓ | |
| PIP3→AKT | ✓ | | | | | | | |
| AKT→ERK | | | ✓ | | ✓ | ✓ | ✓ | |
| AKT→JNK | | | | | | | | ✓ |
| ERK→AKT | | ✓ | | ✓ | ✓ | ✓ | | |
| ERK→PKA | | | | ✓ | | | | |
| PKA→RAF | ✓ | ✓ | | | | | ✓ | ✓ |
| PKA→MEK | ✓ | ✓ | ✓ | ✓ | | ✓ | ✓ | ✓ |
| PKA→ERK | ✓ | ✓ | | | ✓ | | ✓ | ✓ |
| PKA→AKT | ✓ | ✓ | ✓ | ✓ | | ✓ | ✓ | ✓ |
| PKA→PKC | | | | ✓ | | | | |
| PKA→P38 | ✓ | ✓ | ✓ | | | | ✓ | ✓ |
| PKA→JNK | ✓ | ✓ | ✓ | ✓ | | | ✓ | ✓ |
| PKC→RAF | ✓ | ✓ | ✓ | | | | ✓ | ✓ |
| PKC→MEK | ✓ | ✓ | ✓ | ✓ | | | ✓ | ✓ |
| PKC→PLCg | | ✓ | | | | | ✓ | ✓ |
| PKC→PIP2 | | ✓ | | | | | ✓ | ✓ |
| PKC→PIP3 | | | | | | | | ✓ |
| PKC→ERK | | | | | | | ✓ | ✓ |
| PKC→AKT | | | ✓ | | | | ✓ | ✓ |
| PKC→PKA | | ✓ | ✓ | | | | ✓ | |
| PKC→P38 | ✓ | ✓ | ✓ | ✓ | | ✓ | ✓ | ✓ |
| PKC→JNK | ✓ | ✓ | ✓ | ✓ | ✓ | ✓ | ✓ | ✓ |
| P38→JNK | | | | | | ✓ | | |
| P38→PKC | | | | | | ✓ | | |
| JNK→PKC | | | | | | ✓ | | |
| JNK→P38 | | | | ✓ | | ✓ | | |

Table 3: Average execution times for recovering causal relations with different strategies for 2000 models for $n = 6$ variables using the frequentist test with $\alpha = 0.05$.

| Average execution time (s) | | | | |
|---|---|---|---|---|
| Setting | | Direct causal relations | | Only second step | Ancestral relations |
| $n$ | $c$ | ACI with restricted HEJ | direct HEJ | restricted HEJ | ancestral HEJ |
| 6 | 1 | 9.77 | 15.03 | 7.62 | 12.09 |
| 6 | 4 | 16.96 | 314.29 | 14.43 | 432.67 |
| 7 | 1 | 36.13 | 356.49 | 30.68 | 715.74 |
| 8 | 1 | 98.92 | $\geq 2500$ | 81.73 | $\geq 2500$ |
| 9 | 1 | 361.91 | $\geq 2500$ | 240.47 | $\geq 2500$ |

Figure 6: Comparison of ancestral relations predicted by ACI and the score-based method from [9], both using the no-ICAM batch. Depicted are the top 21 ancestral relations obtained by ACI and the transitive closure of the top 17 direct causal relations reported in [9], which results in 21 ancestral relations. Black edges are ancestral relations found by both methods, blue edges were identified only by ACI, while grey edges are present only in the transitive closure of the result from [9].

and asymptotically a reduction of $2^{n^2/4+o(n^2)}$ for $n$ variables. We will refer to this two-step approach as *ACI with restricted HEJ (ACI + HEJ)*. A side effect of assigning infinite scores to the original ancestral predictions instead of the originally estimated scores is that some of the estimated direct causal predictions scores will also be infinite, flattening their ranking. For this preliminary evaluation, we fix this issue by reusing the original ancestral scores also for the infinite direct predictions scores. Another option may be to use the ACI scores for (a)causal relations as soft constraints for HEJ, although at the time of writing it is still unclear whether this would lead to the same speedup as the previously mentioned version.

We compared accuracy and execution times of standard HEJ (without the additional constraints derived from ACI) with ACI with restricted HEJ on simulated data. Figure 7 shows PR curves for predicting the presence and absence of direct causal relations for both methods. In Table 3 we list the execution times for recovering direct causal relations. Additionally, we list the execution times of only the second step of our approach, the *restricted HEJ*, to highlight the improvement in execution time resulting from the restrictions. In this preliminary investigation with simulated data, ACI with restricted HEJ is much faster than standard HEJ (without the additional constraints derived from ACI) for predicting direct causal relations, but only sacrifices a little accuracy (as can be seen in Figure 7). In the last column of Table 3, we show the execution times of standard HEJ when used to score ancestral relations. Interestingly, predicting direct causal relations is faster than predicting ancestral relations with HEJ. Still, for 8 variables the algorithm takes more than 2,500 seconds for all but 6 models of the first 40 simulated models.

Another possible strategy first reconstructs the (possibly incomplete) PAG [12] from ancestral relations and conditional (in)dependences using a procedure similar to LoCI [1], and then recovering direct causal relations. There are some subtleties in the conversion from (possibly incomplete) PAGs to direct causal relations, so we leave this and other PAG based strategies, as well as a better analysis of conversion of ancestral relations to direct causal relations as future work.

## 5 Complete ACI encoding in ASP

Answer Set Programming (ASP) is a widely used declarative programming language based on the stable model semantics of logical programming. A thorough introduction to ASP can be found in [7, 5]. The ASP syntax resembles Prolog, but the computational model is based on the principles

(a) PR direct

(b) PR direct (zoom)

(c) PR direct acausal

Figure 7: Synthetic data: accuracy for the two prediction tasks (direct causal and noncausal relations) for $n = 6$ variables using the frequentist test with $\alpha = 0.05$ for 2000 simulated models.

that have led to faster solvers for propositional logic [7]. ASP has been applied to several NP-hard problems, including learning Bayesian networks and ADMGs [6]. Search problems are reduced to computing the stable models (also called answer sets), which can be optionally scored.

For ACI we use the state-of-the-art ASP solver `clingo 4` [4]. We provide the complete ACI encoding in ASP using the clingo syntax in Table 4. We encode sets via their natural correspondence with binary numbers and use boolean formulas in ASP to encode set-theoretic operations. Since ASP does not support real numbers, we scale all weights by a factor of 1000 and round to the nearest integer.

# 6   Open source code repository

We provide an open-source version of our algorithms and the evaluation framework, which can be easily extended, at `http://github.com/caus-am/aci`.

Table 4: Complete ACI encoding in Answer Set Programming, written in the syntax for clingo 4.

```
%%%%%%%%%%%%%%%%%%%%%%%%%%%%%%%%%%%%%%%%%%%%
%%%%%   Ancestral Causal Inference (ACI)    %%%%%
%%%%%%%%%%%%%%%%%%%%%%%%%%%%%%%%%%%%%%%%%%%%

%%%%% Preliminaries:
%%% Define ancestral structures:
{ causes(X,Y) } :- node(X), node(Y), X!=Y.
:- causes(X,Y), causes(Y,X), node(X), node(Y), X < Y.
:- not causes(X,Z), causes(X,Y), causes(Y,Z), node(X), node(Y), node(Z).

%%% Define the extension of causes to sets.
% existsCauses(Z,W) means there exists I \in W that is caused by Z.
1{causes(Z, I): ismember(W,I)} :- existsCauses(Z,W), node(Z), set(W), not ismember(W,Z).
existsCauses(Z,W) :- causes(Z, I), ismember(W,I), node(I), node(Z), set(W), not ismember(W,Z), Z!=I.

%%% Generate in/dependences in each model based on the input in/dependences.
1{ dep(X,Y,Z);indep(X,Y,Z) }1 :- input_indep(X,Y,Z,_).
1{ dep(X,Y,Z);indep(X,Y,Z) }1 :- input_dep(X,Y,Z,_).

%%% To simplify the rules, add symmetry of in/dependences.
dep(X,Y,Z) :- dep(Y,X,Z), node(X), node(Y), set(Z), X!=Y, not ismember(Z,X), not ismember(Z,Y).
indep(X,Y,Z) :- indep(Y,X,Z), node(X), node(Y), set(Z), X!=Y, not ismember(Z,X), not ismember(Z,Y).

%%%%% Rules from LoCI:
%%% Minimal independence rule (4) : X || Y | W u [Z] => Z -/-> X, Z -/-> Y, Z -/-> W
:- not causes(Z,X), not causes(Z,Y), not existsCauses(Z,W), dep(X,Y,W), indep(X,Y,U),
U==W+2**(Z-1), set(W), node(Z), not ismember(W, Z), Y != Z, X != Z.

%%% Minimal dependence rule (5): X |/| Y | W u [Z] => Z --> X or Z-->Y or Z-->W
:- causes(Z,X), indep(X,Y,W), dep(X,Y,U), U==W+2**(Z-1), set(W), set(U), node(X),
 node(Y), node(Z), not ismember(W, Z), not ismember(W, X), not ismember(W,Y),
 X != Y, Y != Z, X != Z.
% Note: the version with causes(Z,Y) is implied by the symmetry of in/dependences.
:- existsCauses(Z,W), indep(X,Y,W), dep(X,Y,U), U==W+2**(Z-1), set(W), set(U), node(X),
 node(Y), node(Z), not ismember(W, Z), not ismember(W, X), not ismember(W,Y),
 X != Y, Y != Z, X != Z.

%%%%% ACI rules:
%%% Rule 1: X || Y | U and X -/-> U => X -/->Y
:- causes(X,Y), indep(X,Y,U), not existsCauses(X,U), node(X), node(Y), set(U), X != Y,
not ismember(U,X), not ismember(U,Y).

%%% Rule 2: X || Y | W u [Z] => X |/| Z | W
dep(X,Z,W) :- indep(X,Y,W), dep(X,Y,U), U==W+2**(Z-1), set(W), set(U),
node(X), node(Y), node(Z), X != Y, Y != Z, X != Z, not ismember(W,X), not ismember(W,Y).

%%% Rule 3: X |/| Y | W u [Z] => X |/| Z | W
dep(X,Z,W) :- dep(X,Y,W), indep(X,Y,U), U==W+2**(Z-1), set(W), set(U),
node(X), node(Y), node(Z), X != Y, Y != Z, X != Z, not ismember(W,X), not ismember(W,Y).

%%% Rule 4: X || Y | W u [Z] and X || Z | W u U => X || Y | W u U
indep(X,Y,A) :- dep(X,Y,W), indep(X,Y,U), U==W+2**(Z-1), indep(X,Z,A), A==W+2**(B-1),
 set(W), set(U), not ismember(W,X), not ismember(W,Y), node(X), node(Y), node(Z),
 set(A), node(B), X!=B, Y!=B, Z!=B, X != Y, Y != Z, X != Z.

%%% Rule 5: Z |/| X | W and Z |/| Y | W and X || Y | W => X |/| Z | W u Z
dep(X,Y,U) :- dep(Z,X,W), dep(Z,Y,W), indep(X,Y,W), node(X), node(Y), U==W+2**(Z-1),
 set(W), set(U), X != Y, Y != Z, X != Z, not ismember(W,X), not ismember(W,Y).

%%%%% Loss function and optimization.
%%% Define the loss function as the incongruence between the input in/dependences
%%% and the in/dependences of the model.
fail(X,Y,Z,W) :- dep(X,Y,Z), input_indep(X,Y,Z,W).
fail(X,Y,Z,W) :- indep(X,Y,Z), input_dep(X,Y,Z,W).

%%% Include the weighted ancestral relations in the loss function.
fail(X,Y,-1,W) :- causes(X,Y), wnotcauses(X,Y,W), node(X), node(Y), X != Y.
fail(X,Y,-1,W) :- not causes(X,Y), wcauses(X,Y,W), node(X), node(Y), X != Y.

%%% Optimization part: minimize the sum of W of all fail predicates that are true.
#minimize{W,X,Y,C:fail(X,Y,C,W) }.
```