[Reviews · NeurIPS 2016]

Reviewer 1

Summary

The paper presents a hybrid (constraint-based and score-based) approach for causal discovery called Ancestral Causal Inference (ACI). ACI is inspired by a previous work (Hyttinen et al., 2014) which uses global optimization to learn a causal graph. The authors suggest a coarse-grained representation of the causal structure based on ancestral relations to address the scalability issue in global optimization. They present a set of ancestral reasoning rules and novel causal reasoning rules and propose a method to score the output causal predictions. ACI takes an input a set of weighted (in)dependence statements and ancestral relations and models the task of causal discovery as an optimization problem subject to the reasoning rules. The authors evaluate their approach on synthetic and real world data and show that ACI outperforms the state-of-the-art methods.

Qualitative Assessment

The paper is well written with a clear presentation of the motivation, proposed method, and evaluation. However, I would like to see the following points clarified / addressed in their review: 1- In the analysis of the evaluation over synthetic data, the authors say that ACI is performing "significantly better for nonancestral predictions and the top ancestral predictions". However, the PR graphs of the non-ancestral predictions show that the bootstrapped FCI and CFCI perform better for a high recall subject to a minor decrease in precision. A similar observation is present in the PR graphs of ancestral prediction but the loss in precision is significant. It should be justified by the authors if the precision is more crucial than the recall. 2- In the introduction lines 44-45: "we only represent ancestral relations". It would be more clear to present a brief explanation (2-3 statements) of the alternative. Addressing this issue clarifies the numerical comparison made in the third section lines 125-126 about the improvement in the search space over previous related work. 3- The authors mention the Bayesian weights as part of the experimental conditions over the synthetic data, however, the results are not discussed in the paper. The results are present in the supplementary material but it is better to reference them or provide a brief summary in the paper. 4- Reference [18] is broken: "J. Ramsey, J. Zhang, and P. Spirtes. In UAI, 2006."

Confidence in this Review

2-Confident (read it all; understood it all reasonably well)


Reviewer 2

Summary

The authors study the problem of causal discovery from limited data and propose a hybrid (both constraint and score based) approach called ACI. This approach solves a constraint optimization problem to discover the ancestral causal relationship of a system. Using the ancestral causal representation, they could reduce the computation time compared to the other approaches such as the one in [10]. The proposed method scores the causal predictions based on their confidence, which allows the predictor to rank his predictions according to their reliability. Moreover, ACI allows its user to combine observational and interventional data in order to improve the precision. The authors prove soundness and asymptotic consistency of their method and demonstrate its performance on synthetic and real world datasets.

Qualitative Assessment

Strengths: - This paper studies an interesting and practical problem. It is well written and easy to follow. - Recovering the ancestral graph instead of DAG reduces the search space and improves the computation time. - Scoring the causal prediction is interesting and allows the user to rank the predictions according to their reliability. - Although the precision of the proposed method (ACI) and the method proposed in [10] are quite similar, its computation time looks drastically better, which is also advocated by the simulation result and Figure 1. Weaknesses: - The proposed method is very similar in spirit to the approach in [10]. It seems that the method in [10] can also be equipped with scoring causal predictions and the interventional data. If otherwise, why [10] cannot use these side information? - The proposed method reduces the computation time drastically compared to [10] but this is achieved by reducing the search space to the ancestral graphs. This means that the output of ACI has less information compared to the output of [10] that has a richer search space, i.e., DAGs. This is the price that has been paid to gain a better performance. How much information of a DAG is encoded in its corresponding ancestral graph? - Second rule in Lemma 2, i.e., Eq (7) and the definition of minimal conditional dependence seem to be conflicting. Taking Z’ in this definition to be the empty set, we should have that x and y are independent given W, but Eq. (7) says otherwise.

Confidence in this Review

2-Confident (read it all; understood it all reasonably well)


Reviewer 3

Summary

This paper introduces an algorithm that learns causal ancestral structures from conditional independence tests. The test results are weighted based on their perceived reliability and the method uses answer set programming to find the structure that minimizes the sum of the weights of violated constraints. The experiments show that the method is both accurate and relatively fast.

Qualitative Assessment

This work contributes a new algorithm that tries to handle errors in conditional independence tests. The algorithm improves state-of-the-art both in terms of accuracy and speed. Thus, there is potential for a significant impact. Overall, the paper was well-written and easy to follow. There seems to be an error in rule (7) in Lemma 2. By the definition of minimal conditional dependence, X is conditionally independent of Z given W. Fonts in the figures are too small. Especially, Figure 2 is very hard to comprehend because it is difficult to figure out which curve corresponds to which method. Further, yellow is a bad choice for the color of a curve. To put things in perspective, it would be nice to know how the running time of FCI and CFCI compares to the current method. Reference [18] is missing a title.

Confidence in this Review

2-Confident (read it all; understood it all reasonably well)


Reviewer 4

Summary

This paper formulates the causal discovery problem as an optimization problem and thereby, proposes a sound and (computationally) efficient method for reconstructing ancestral structures. To this end, the paper proposes 6 new causal reasoning rules (in lemma-2) and a sound scoring method. The methods proposed here are applicable in the presence of latent variables as well.

Qualitative Assessment

Causal discovery is a hard problem and this paper proposes a reasonably efficient and seemingly easy to implement solution to this problem. Interestingly, the solution proposed here also works in the presence of latent variables. The part dealing with combining observational and interventional data is not clear to me. What exactly do the authors mean by "suitable" interventional data? Noting that the authors consider this as a crucial contribution (stated in the abstract), it will be very useful if they can explicate this section in a better manner. How do you bring down the combinatorial explosion? Although you mention this on the last paragraph in page-3, you do not talk about this further in the (theory section of the) paper. To what degree is your method helpful in reducing or containing the complexity?

Confidence in this Review

2-Confident (read it all; understood it all reasonably well)


Reviewer 5

Summary

This paper considers learning a causal structure through constraint testing, representing that structure as an 'ancestral' object rather than the more common directed acyclic graphs. I think this is essentially equivalent to representing each DAG by its transitive closure (perhaps the authors could comment on this).

Qualitative Assessment

Overall this is a nice method to an important problem. The setup seems quite flexible, so I can imagine incorporating other kinds of information relatively easily. The use of ancestral models rather than Bayesian networks is sensible and novel to my knowledge - though see below for a comment about recovering a BN. The paper is well written, seems technically solid and novel, and provides a method that could be very useful. Some questions and suggestions for improvement are given below. ---- COMMENTS Page 2 line 46 - could you be a bit more precise when saying 'keep the computational burden low'. Relatedly, it would be helpful to discuss how quickly the number of ancestral objects grows asymptotically, and whether Chickering's result about learning BNs being NP-hard applies to ancestral objects as well (is this is easy to check). Lemma 2 - it would be useful to have here a signpost to the discussion after Theorem 1. The list of rules immediately had me wondering: "how did they come up with this list and is it complete?" In the simulations, I think it's a bit unreasonable to throw in the red curve (which simulates what happens if some interventional data are available), since none of the other methods claim to be able to deal with this. This should be removed from the plot, as it is misleading. What's going on with CFCI and FCI in the plots - they seem to have essentially no points on the plot. Is this because it's too computationally hard to run FCI with different test sizes? If so, perhaps using RFCI instead would help (I'm not sure if there is a CRFCI though!). The description of the bootstrapping is a bit hard to follow - is this somewhat like stability selection? Perhaps you could use the supplementary material to explain this better. For the interventional setting, the method of Peters, Buhlmann and Meinshausen (2016) [invariant prediction] would seem like the most sensible comparison. It can deal with a very flexible class of interventions, so would make a good counterpoint to the discussion at the end of Section 3. Given the ancestral object, is it hard then just to recover a directed graph by doing some additional tests? Perhaps this is harder than it seems because parts of the graph may not have an ordering at all, but I'd be interested to know the authors' thoughts. ---- TYPOS ETC Though not intentional, a naive reading of the second paragraph of the introduction gives the impression that 'hybrid methods' are a creation of this paper! There are, of course, many other hybrid methods available for learning Bayesian networks and therefore causal structures. p5 line 191 - "allow to deduce" -> "allow one to deduce" Figure 3 means nothing to me - what is it supposed to represent? Could you give the ancestral structure discovered in the appendix? It would be good to compare to the 'ground truth' and the Sachs method.

Confidence in this Review

3-Expert (read the paper in detail, know the area, quite certain of my opinion)


Reviewer 6

Summary

This paper introduces Ancestral Causal Inference aiming to recover ancestral relations (X being ancestor of Y) instead of the full fine-grained causal structure. Providing the necessary theoretical foundations and inference rules, the authors present an algorithm to perform inference using answer set programming and discuss how to score the obtained causal predictions. The method is compared to (bootstrapped) (C)FCI and COMBINE on simulated data and a real-world protein data set.

Qualitative Assessment

This manuscript is written to a very high standard and the organisation and line of argument is good throughout the manuscript. It lays out interesting ideas and concepts that inspire and allow future research building upon the defined concepts. It may be fruitful to further discuss how results of ACI can be readily extended in order to obtain more fine-grained causal insights. While the authors are right in stressing the computational benefit of their methods it is important to address that one looses on detail of causal insight (it is a trade-off and not a win-win situation). This is why I view it important to further discuss this aspect (cf. g) below). a) Please ensure all abbreviations/acronyms are introduced at first occurence (e.g. PC/FCI in Line 18, (C)FCI in Line 69, CFCI and BCCD and COMBINE in Line 104, ADMGs in Line 144, CFCI in Line 233). b) Line 20: Please consider replacing "significant" by "substantial" or another synonymous word. c) Line 82: Please define "causal relationship" at this point in the manuscript to increase accessibility to a broader audience. d) Typo in Line 151: Missing space after "semantics". e) Figure 1(b) and Figure 2: Please change the font to match the serif font of the manuscript, increase the font size, and increase the line width for clarity. f) Line 246: "going on" is a little colloquial ;-) g) Line 284: Please elaborate on this point and clarify _how_ this can be done since it is an important point to discuss to further convince readers of the practicality and use of the proposed ACI method.

Confidence in this Review

2-Confident (read it all; understood it all reasonably well)